# Leveraging Function Space Aggregation for Federated Learning at Scale

**Nikita Dhawan**[*][†]                                                 *nikita@cs.toronto.edu*
*University of Toronto and Vector Institute*

**Nicole Mitchell**[*]                                                 *nicolemitchell@google.com*
*Google Research*

**Zachary Charles**                                                 *zachcharles@google.com*
*Google Research*

**Zachary Garrett**                                                 *zachgarrett@google.com*
*Google Research*

**Gintare Karolina Dziugaite**                                                 *gkdz@google.com*
*Google DeepMind and Mila - Quebec AI Institute*

**Reviewed on OpenReview:** *https://openreview.net/forum?id=Ytp9KFKZfZ*

## Abstract

The federated learning paradigm has motivated the development of methods for aggregating multiple client updates into a global server model, without sharing client data. Many federated learning algorithms, including the canonical Federated Averaging (FEDAVG), take a direct (possibly weighted) average of the client parameter updates, motivated by results in distributed optimization. In this work, we adopt a function space perspective and propose a new algorithm, FEDFISH, that aggregates local approximations to the functions learned by clients, using an estimate based on their Fisher information. We evaluate FEDFISH on realistic, large-scale cross-device benchmarks. While the performance of FEDAVG can suffer as client models drift further apart, we demonstrate that FEDFISH is more robust to longer local training. Our evaluation across several settings in image and language benchmarks shows that FEDFISH outperforms FEDAVG as local training epochs increase. Further, FED-FISH results in global networks that are more amenable to efficient personalization via local fine-tuning on the same or shifted data distributions. For instance, federated pretraining on the C4 dataset, followed by few-shot personalization on Stack Overflow, results in a 7% improvement in next-token prediction by FEDFISH over FEDAVG.

## 1 Introduction

Methods for aggregating separately trained neural networks have received renewed attention as machine learning models and data reach ever larger scales (Wortsman et al., 2022; Li et al., 2022; Rame et al., 2023). Parallel training can yield gains in computational efficiency (Assran et al., 2020; Li et al., 2022) and meet constraints on access to private data, as in Federated Learning (FL; McMahan et al., 2017). Model aggregation plays a central role in how clients collaboratively train a model in a distributed manner via FL without sharing data among each other or with any orchestrating server. Given that the *cross-device* FL setting is characterized by client data heterogeneity, unreliable client availability and network constraints (Kairouz et al., 2021), FL is typically carried out over multiple communication rounds in which updates from

---

[*]Equal contribution.
[†]Work done as Student Researcher at Google Research.

local training are aggregated to iteratively improve the global model. The canonical approach to aggregation, implemented by the FEDAVG method and its adaptive variants (Reddi et al., 2020), is to combine client model parameter updates by averaging them, weighted in proportion to their respective dataset sizes.

In this work, we take a function space perspective (Benjamin et al., 2018) of model aggregation in FL, where we aim to obtain a global model that simultaneously matches each client model's logit outputs on that client's data. One motivation for this is to allow for, and reap the benefits of, more local training between global updates. The performance of algorithms like FEDAVG depends heavily on the number of local training iterations. When the data are heterogeneous across clients, training too long between communication rounds leads to updates that hurt the global model's performance, a phenomenon known as client drift (Karimireddy et al., 2020). Indeed, prior work has shown that the number of local training steps dictates a trade-off between the speed of convergence and the quality of the resulting model (Charles & Konečnỳ, 2021; Malinovskiy et al., 2020; Pathak & Wainwright, 2020). These results imply that selecting the number of local steps is critical. It is also challenging, in part, due to the difficulties of hyperparameter tuning in federated settings (Kuo et al., 2023), and in part because the number of local steps has wide ranging effects. These include not only the speed of convergence (Pathak & Wainwright, 2020; Mitra et al., 2021), but also the optimization dynamics (Charles & Rush, 2022) and even whether the method acts as a meta-learner (Collins et al., 2022; Charles et al., 2023). While there are a variety of FL methods aimed at mitigating client drift (see Wang et al. (2021) for an overview), many of these introduce extra hyperparameters to tune and remain sensitive to the number of local steps (Mitra et al., 2021; Charles & Konečnỳ, 2021). Instead, we argue that the choice of model aggregation technique plays a role in the client drift problem. Taking a function space view of the client models sidesteps the drift problem by aiming for a global model that in the function space more accurately represents each client model.

A key obstacle to our function space approach, matching the client models' outputs on client data, is its dependence on client data. FL constrains access to client data, preventing a direct approach where the global server uses client data to match the corresponding model outputs. As a step towards parametric function space model aggregation which does not require direct access to client data, we propose and implement a Fisher-weighted federated averaging algorithm, called FEDFISH. This method is derived from an objective that minimizes an approximate function space distance between local client models and the global model. The closed-form solution of this objective depends on the networks' Fisher Information (Cover, 1999), which are typically too expensive to compute and store for large models. The approximations required to implement our method in practice lead to a parametric aggregation scheme which accounts for the client data distributions via the functions represented by their local models. We investigate the advantage this confers upon FEDFISH over simple averaging (FEDAVG) in regression, image classification and language modeling benchmarks.

Our extensive evaluation includes domain-specific criteria as well as metrics specific to FL. We demonstrate settings in which FEDFISH outperforms FEDAVG, especially as the amount of local training is varied. Image and language experiments with varying levels of client data heterogeneity show improved post-personalization performance of FEDFISH throughout training, when the global model is locally fine-tuned for a few steps by clients that were held out during training. This observation also holds when measuring transfer performance by drawing the evaluation clients from a shifted data distribution. For instance, in an experiment with federated pretraining on the large and hetergenous C4 dataset, followed by few-shot personalization on Stack Overflow clients, FEDFISH is able to improve upon FEDAVG's next-token prediction performance by 5-7%, depending on the amount of personalization data available. We provide insight into these gains by assessing a measure of deviation between global and local models, coined the Client-Server Barrier. Finally, we discuss the impact of these methods and settings on the cost of communication between clients and the server.

**Contributions.**

- We formalize a function space perspective of federated learning to motivate a scalable algorithm, FEDFISH, which aims to match client input–output functions during aggregation.

- Via a synthetic example, we demonstrate that FEDFISH outperforms FEDAVG as client data heterogeneity increases. We then investigate this performance at larger scales than have been explored by previous works.

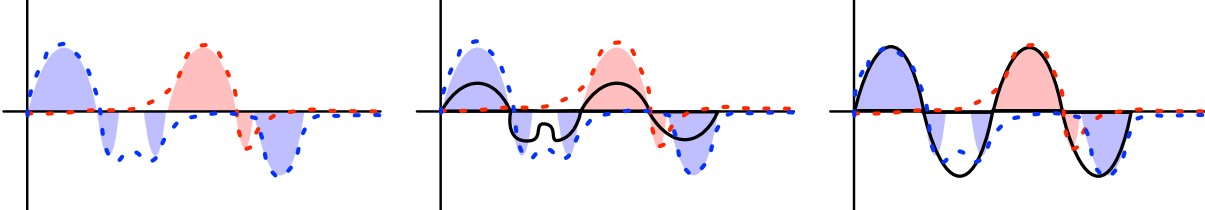

Figure 1: Given two functions modeled over disjoint supports (**left**), a direct parameter average fails to represent either function well (**center**), while function space aggregation aims to preserve both functional relationships (**right**).

- Our thorough empirical results show that FEDFISH allows for longer local client training compared to FEDAVG. We find that the global models learned via FEDFISH have greater ability to be personalized via fine-tuning on the same or shifted data distributions, indicating they provide a better initialization for local training in each round.

- We propose to evaluate effects of aggregation via a Client-Server Barrier, leveraging the function space perspective to gain further insight into the observed results.

## 2 Federated Learning in the Function Space

We now define the federated learning problem from a function space perspective and describe the approximations that lead to a practical and parametric objective.

### 2.1 Problem Setting

Let a network parameterized by $\theta$ be trained on a dataset of input–target pairs, $D = (\mathbf{X}, \mathbf{y})$, to optimize a loss function $\mathcal{L}$, such that it represents a function $f(\mathbf{X}; \theta) = \mathbf{Z}$, where $\mathbf{Z}$ denotes the network's outputs across all inputs. Consider the canonical federated learning setting, where a global model's parameters $\theta_G$ are broadcast to $N$ clients for local training. Each client $i$ trains their model on local data $D_i$ (with a corresponding set of input data $\mathbf{X}_i$) for a fixed number of iterations to produce trained parameters $\theta_i$. These parameters are then communicated back to a global server where they are aggregated using a specific aggregation technique. This procedure is repeated over multiple rounds, where the aggregated model from each round serves as the initialization for local training in the subsequent round.

Viewing FL from a function space perspective, the aggregated model should ideally match the input–output relationships learned by each client so far. More formally, for each federated round, we define the *optimal global model* as the one that produces outputs closest to each client's outputs when evaluated on the corresponding input data. Let us denote this function space distance as $\mathcal{D}(\cdot, \cdot)$. Averaging this quantity over all clients, we obtain the following objective:

$$\theta_G^* = \arg\min_\theta \frac{1}{N} \sum_{i=1}^{N} \mathcal{D}\left(f(\mathbf{X}_i; \theta), f(\mathbf{X}_i; \theta_i)\right). \tag{1}$$

We depict this idealized objective in fig. 1, where two client functions are learned on different supports (left) and we wish to aggregate them into a model that preserves both functional relationships (right), which direct parameter averaging cannot achieve (center).

The objective in eq. (1) depends on function outputs, which in turn rely on client-specific inputs $\mathbf{X}_i$. Exactly implementing this would require global access to the local client data, which violates a fundamental constraint in federated learning. This necessitates a parametric approximation to the function space distance such that it may be estimated without actual data points.

## 2.2 Approximating Function Space Distance

The function space distance can be estimated with a second-order Taylor approximation with respect to model parameters $\theta$, centered at $\theta_i$, the client network whose outputs are to be matched. This is useful in a federated context because appropriate approximations to this estimate lead to a parametric method which does not directly depend on client data.

Setting $\mathcal{D}(\cdot, \cdot)$ to be the Kullback-Leibler (KL) divergence between softmax outputs of the networks,

$$\mathcal{D}(f(\mathbf{X}_i; \theta), f(\mathbf{X}_i; \theta_i)) \approx \frac{1}{2}(\theta - \theta_i)^T F_i(\theta - \theta_i) \tag{2}$$

$$\approx \frac{1}{2} \sum_{j=1}^{|\theta_i|} F_i^{(j)} (\theta^{(j)} - \theta_i^{(j)})^2, \tag{3}$$

where $F_i$ is the Fisher Information matrix corresponding to $\theta_i$. In eq. (2), the zero-th and first order terms vanish because $\mathcal{D}(\cdot, \cdot)$ is a non-negative function that evaluates to zero when its arguments are equal. Hence, its value and gradient both vanish at $\theta = \theta_i$, leaving only the second order term. Note that this approximation does not require $\theta_i$ to be optimal and can be used at intermediate stages of training in multi-round FL. We defer complete details of this derivation to appendix A.2.

The full Fisher Information matrix is expensive to compute and store for large scale networks. Further, the corresponding closed-form solution to this optimization problem would involve the inverse sum of Fisher Information matrices, which need not be invertible in practice. Common approximations (Kirkpatrick et al., 2017) involve using the diagonal empirical Fisher Information matrix, as shown in eq. (3), where $F_i^{(j)}$ is the $j$-th diagonal entry of $F_i$.

## 3 FedFish Algorithm

Given function space distance approximations in section 2.2, we now obtain a parametric aggregation scheme that can be practically implemented in federated settings. Plugging eq. (3) into eq. (1) and solving the convex optimization problem gives:

$$\theta_G^* = \arg\min_\theta \frac{1}{2N} \sum_{i=1}^N \sum_{j=1}^{|\theta_i|} F_i^{(j)} (\theta^{(j)} - \theta_i^{(j)})^2 = \frac{\sum_{i=1}^N \text{diag}(F_i)^T \theta_i}{\sum_{i=1}^N \text{diag}(F_i)}, \tag{4}$$

where $\text{diag}(F_i)$ represents the diagonal of $F_i$. Hence, the global model at each round is the Fisher-weighted average of client models, normalized by the sum of all Fisher diagonals. In this form, FEDFISH is simple to implement and efficient to deploy in cross-device federated settings. The empirical Fisher diagonal can be computed during local training, via an average of squared gradients of the training objective with respect to mini-batches of data, as shown in algorithm 2. Similar to common FEDAVG implementations, the clients communicate model deltas to the global server, which are then aggregated and used as pseudo-gradients in one step of global model optimization. FEDFISH can be combined with adaptive optimization techniques for different choices of global optimizers. We show this procedure for SGD with global learning rate $\eta_g$ and local learning rate $\eta_c$ in algorithm 1. See appendix A.5 for a discussion of efficiency improvements.

Note that we recover the FEDAVG algorithm if the Fisher coefficients for each parameter are the same across client networks. Such a condition is unlikely to hold even approximately, especially in settings with increasing data heterogeneity. Fisher coefficients account for the influence of each parameter on its network's predictions with respect to corresponding inputs. Intuitively, FEDFISH has the following advantages: (1) Each local parameter's contribution to the global aggregate is proportionate to a measure of its importance for making predictions on its training data, which improves upon the FEDAVG global model. (2) Derived from a function space perspective, FEDFISH simply aims to match the functions learned by clients so far, without relying on optimality assumptions. This makes it compatible with heterogeneous datasets, longer local training as well as multiple rounds of FL. We empirically test for these gains across several settings ranging from toy to large scales.

**Algorithm 1** FEDFISH (SGD)

---

**Require:** rounds $R$, local epochs $E$, $\theta_G$, client datasets $D$, global lr $\eta_g$, local lr $\eta_c$

1: **for** $r \leftarrow 1$ to $R$ **do**
2:    Sample a cohort of clients, $C$
3:    **for** $i \in C$ **in parallel do**
4:      Compute client weights, $w_i = |D_i|$
5:      $\theta_i \leftarrow \theta_G$
6:      $\Delta\theta_i, F_i \leftarrow \texttt{FedLocalTrain}(E, \theta_i, D_i, \eta_c)$
7:    **end for**
8:    $\theta_G \leftarrow \theta_G - \eta_g \dfrac{\sum_{i=1}^{N} w_i F_i^T \Delta\theta_i}{\sum_{i=1}^{N} w_i F_i}$
9: **end for**
10: **return** $\theta_G$

**Algorithm 2** FedLocalTrain (SGD)

---

**Require:** $E, \theta_i, D_i, \eta_c$

1: $\texttt{ModelDelta}, \texttt{SumFisher} \leftarrow 0, 0$
2: **for** $e \leftarrow 1$ to $E$ **do**
3:    **for** $b \in D_i$ **do**
4:      $g \leftarrow \nabla_\theta \mathcal{L}(\theta_i, b)$
5:      $\theta_i \leftarrow \theta_i - \eta_c\, g$
6:      $\texttt{ModelDelta} \leftarrow \texttt{ModelDelta} + \eta_c\, g$
7:    **end for**
8: **end for**
9: **for** $b \in D_i$ **do**
10:    $g \leftarrow \nabla_\theta \mathcal{L}(\theta_i, b)$
11:    $\texttt{SumFisher} \leftarrow \texttt{SumFisher} + g^2$
12: **end for**
13: **return** $\texttt{ModelDelta}, \texttt{SumFisher}$

## 4 Evaluation and Client-Server Barrier

While there are natural domain-relevant evaluation metrics for our benchmarks, here, we describe specific criterion relevant to FL. The commonly used global and personalization performance are useful indicators of successful FL algorithms. However, they may be confounded by local optimization choices. Hence, we formalize the Client-Server Barrier below and additionally evaluate it in section 5, as a more direct measure of the quality of a given aggregation method. The criteria below are defined in terms of a performance metric $L_i(\cdot)$ that measures a quantity of interest, such as loss or prediction error, on client data $D_i$.

**Global Performance.** The most natural measure of success in FL is the performance of the global model on held-out client data, averaged over clients. Using our notation from before, this is given by $\frac{1}{N}\sum_{i=1}^{N} L_i(\theta_G)$.

**Client Personalization Performance.** While global performance is akin to a network's zero-shot abilities, we are often interested in its personalization ability to unseen clients after a few steps of fine-tuning. Quick adaptation of the network to particular clients or downstream use cases is critical in the compute-constrained settings that FL targets. We measure client personalization by fine-tuning $\theta_G$ on a portion of each held-out client dataset and then evaluating each fine-tuned model on the remaining unseen client data (the same portion used for measuring global performance), averaging over clients as before.

**Client-Server Barrier.** For a given performance metric $L$ and aggregation technique, we define the Client-Server Barrier as the difference in this performance metric between each client model $\theta_i$ and the aggregated global model $\theta_G$ with respect to the client data $D_i$, averaged over all the clients involved in the aggregation. Mathematically, this is given by

$$\frac{1}{N}\sum_{i=1}^{N}\left(L_i(\theta_G) - L_i(\theta_i)\right) = \frac{1}{N}\sum_{i=1}^{N} L_i(\theta_G) - \frac{1}{N}\sum_{i=1}^{N} L_i(\theta_i). \tag{5}$$

This is a simple and direct measure of the impact of aggregation on model performance and can be computed in a federated manner: averaging the broadcast global model's performance across all client data in the sampled cohort (first term), and averaging trained local models' performance on their respective client data across the sampled cohort (second term).

## 5 Experiments

Using the criteria described in section 4, we now conduct a systematic empirical evaluation of FEDFISH in varied settings, compared to the best performing variant of FEDAVG. We first demonstrate the advantage of FEDFISH as client data heterogeneity increases in a toy regression problem. We then assess its performance across settings in larger scale image and language benchmarks.

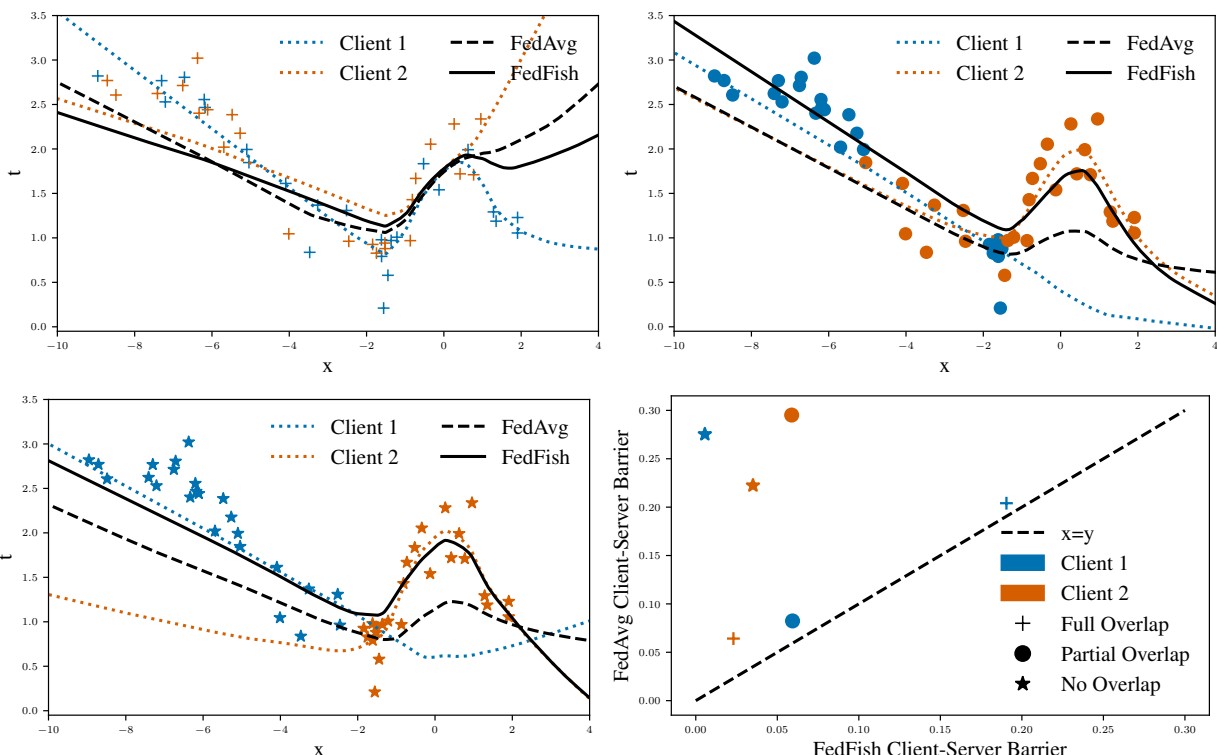

Figure 2: As heterogeneity across clients increases (**top left → top right → bottom left**), FedAvg deteriorates, while FedFish matches predictions of both client models. For each setting shown and each client within it, the FedFish global model has lower barrier to the clients (**bottom right**).

## 5.1 A Toy Regression Demonstration

Figure 2 shows a non-linear regression problem with two clients across which data is distributed with varying heterogeneity, including full ($+$), partial ($\circ$) and no overlap ($\star$). We plot the local functions learned by each client, as well as the global functions produced by aggregation via FedAvg and FedFish after one round. For completely homogeneous client data, FedAvg and FedFish fit similar functions. When there is partial overlap, FedAvg seems to reasonably retain predictions on one client's data while poorly fitting the other, while FedFish fits both datasets well. In the extreme case of completely disjoint supports, FedAvg fails to fit either client dataset, but FedFish matches the locally learned functions of both clients on their respective input data. The Client-Server Barrier (CSB) defined in eq. (5) is computed in terms of mean squared error for each client on their corresponding data. As shown by all the points above the $x = y$ line in fig. 2 (bottom right), the CSB is lower for FedFish than FedAvg in each of these settings, with more significant difference as data heterogeneity increases. We hypothesize that accounting for the functions learned by local models confers this advantage upon FedFish.

## 5.2 Image Classification and Text Benchmarks

**Datasets and architectures.** We consider a variety of federated benchmarks for image classification (EMNIST (Cohen et al., 2017), CIFAR100 (Krizhevsky et al.)) and language modeling (Stack Overflow (Authors, 2019), CC-News (Hamborg et al., 2017) and C4 (Raffel et al., 2020)). In particular, C4 is a large-scale and significantly heterogenous dataset. For these domains, we use standard classifier and transformer architectures, respectively.

For EMNIST, we partition the handwritten characters according to their author, as proposed by Caldas et al. (2018). For Stack Overflow, posts on the eponymous web forum are partitioned by their author as well. For CIFAR100, we partition the examples over 100 clients in a heterogeneous fashion using the

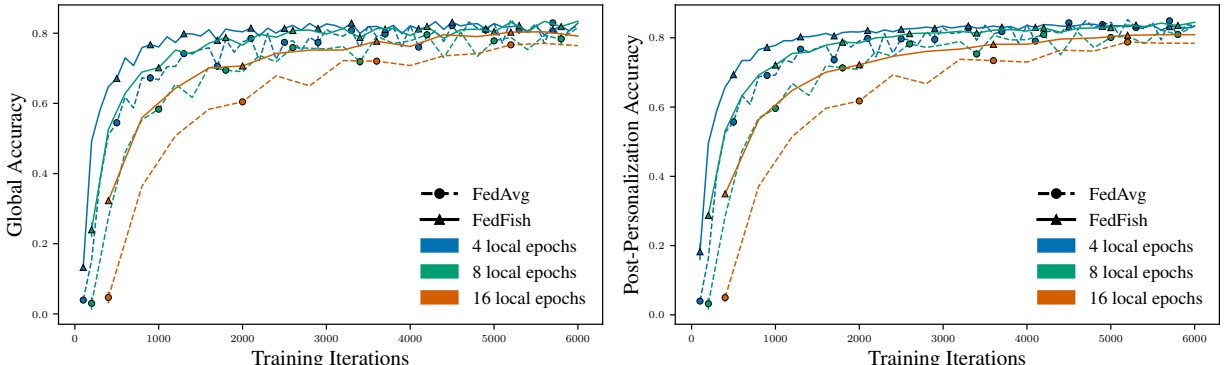

Figure 3: Training on EMNIST with FEDFISH converges faster and to a higher global accuracy (**left**) and post-personalization accuracy (**right**) than training with FEDAVG, across varying numbers of local epochs. Results are shown with fixed compute across configurations: each training iteration corresponds to a local epoch, and each marker indicates 100 federated communication rounds.

two-level latent Dirichlet allocation scheme proposed by Reddi et al. (2020). For CC-News and C4, we use Dataset Grouper (Charles et al., 2023) to partition web-crawled examples according to their base URL (e.g. `nytimes.com`). More details about data splits, architectures and hyperparameters are included in appendix A.3.

**Performance metrics.** We evaluate global performance, client personalization performance and Client-Server Barrier (see section 4), using standard domain-relevant performance metrics. These include classification accuracy for images and next-token prediction accuracy and perplexity for language modeling. Since C4 is a very large scale dataset that may generally be used as a pretraining corpus, we evaluate its global model on held-out clients from C4 itself as well as on the shifted Stack Overflow and CC-News datasets. This tests the methods' transfer performance in addition to adaptability to new clients that were not seen during training. For the C4 experiments, we also vary the amount of fine-tuning data used for personalization – 25% or 50% of each held-out client's data – to assess few-shot performance. Additional details are reported in A.3.

### 5.2.1 Effect of Local Training on Global Model Performance

We study the effect of local training on the global model's performance by varying the number of epochs of training clients perform in between rounds. Since increasing the number of local epochs for a fixed number of rounds increases computational costs, we present our results by separately fixing compute (or total number of training iterations) and number of aggregation rounds. We provide a complete table of results covering all settings in table 3 of appendix A.4 and discuss representative experiments here. With fixed compute, fig. 3 (left) shows a decline in global accuracy as number of local epochs is increased for both FEDAVG and FEDFISH, as expected by the client-drift phenomenon. However, within each setting, and across datasets, FEDFISH outperforms FEDAVG, suffering a more graceful decline in performance with more local training and converging to higher performance faster. Figure 4 similarly shows FEDFISH outperforming FEDAVG in terms of classification accuracy and next-token prediction accuracy for CIFAR100 (left) and Stack Overflow (right) datasets, respectively. In both of these settings, FEDFISH is much more robust to longer periods of local training that FEDAVG, whose performance suffers as local training increases. In fig. 5 (left), we use the heterogeneous C4 dataset for federated training and evaluate its zero-shot performance on unseen clients from C4 itself. The bottom row of the plot shows that the global model, without any personalization data, benefits considerably from increased local epochs during federated pretraining for both algorithms, with FEDFISH outperforming FEDAVG.

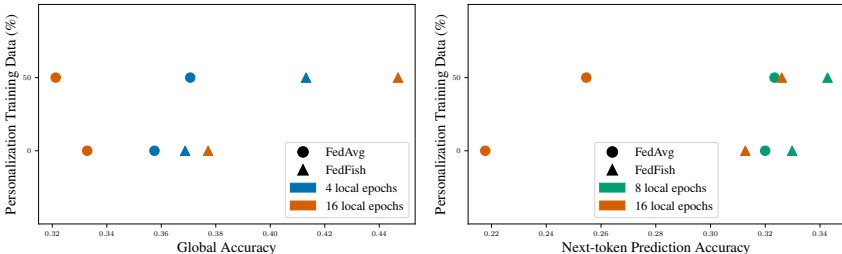

Figure 4: Global and post-personalization performance in terms of classification accuracy on CIFAR100 (**left**) and next-token prediction accuracy on Stack Overflow (**right**). Varying number of local training epochs can significantly impact FEDAVG performance while FEDFISH remains relatively robust to this.

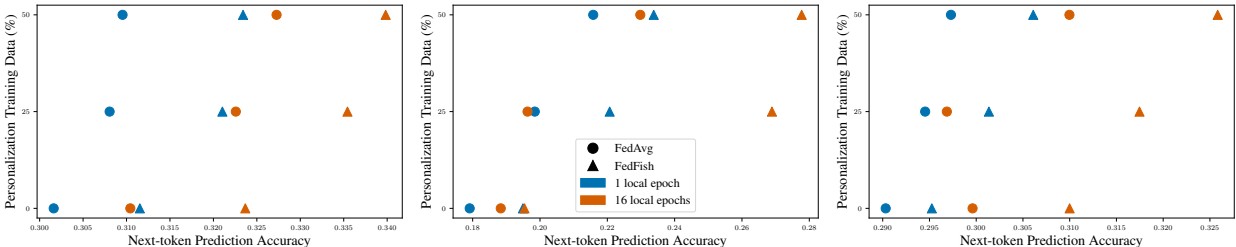

Figure 5: Transfer (global and post-personalization) performance in terms of next-token prediction after federated pretraining on C4 and evaluating on C4 (**left**), Stack Overflow (**center**) and CC-News (**right**). Personalizing with 0%, 25% or 50% of held-out client data results in FEDFISH outperforming FEDAVG, especially with longer local training.

### 5.2.2 Post-Personalization Performance

At scale, pretrained models are often personalized or fine-tuned for a small number of steps using local client data before deployment. Accordingly, we fine-tune the global models for a few steps on limited datapoints from held-out clients and evaluate the metrics discussed earlier. Consistent with global performance reported above, we observe across tasks that FEDFISH yields models with higher post-personalization performance than those trained with FEDAVG. This is shown on EMNIST in fig. 3 (right), on CIFAR100 in fig. 4 (left), on Stack Overflow in fig. 4 (right), and on C4 in fig. 5 (left). Notably we see that personalization worsens performance on CIFAR100 trained with FEDAVG using 16 local epochs, while substantially improving CIFAR100 trained with FEDFISH using the same configuration. By contrast, FEDAVG with 16 local epochs on Stack Overflow improves dramatically with personalization, despite still underperforming all other models. Interestingly, we see in the case of C4 that more than aggregation algorithm the amount of local training seems to impact personalization performance. While both FEDFISH models trained with 1 or 16 local epochs have higher zero-shot performance than either FEDAVG model, the 16 local epoch FEDAVG model's post-personalization performance surpasses that of FEDFISH with 1 local epoch as the amount of fine-tuning data increases. Overall, we find that models trained with FEDFISH using longer periods of local training tend to be more amenable to personalization than models trained with FEDAVG.

### 5.2.3 Transfer Performance

Considering FEDAVG and FEDFISH as methods of federated pretraining, we further evaluate the networks trained on C4 in terms of their transfer performance (fig. 5) on Stack Overflow (center) and CC-News (right). Here, performance is in terms of next-token prediction accuracy. We similarly report the perplexity for each of these settings in appendix A.3.3. We vary the amount of data available for personalization, between 0%, 25% and 50% of each held-out client dataset. The reported performance is always evaluated on the unseen 50% of the data. In each of these settings, we find that transfer performance benefits from longer local training for both methods and FEDFISH yields better zero-shot, few-shot and post-personalization

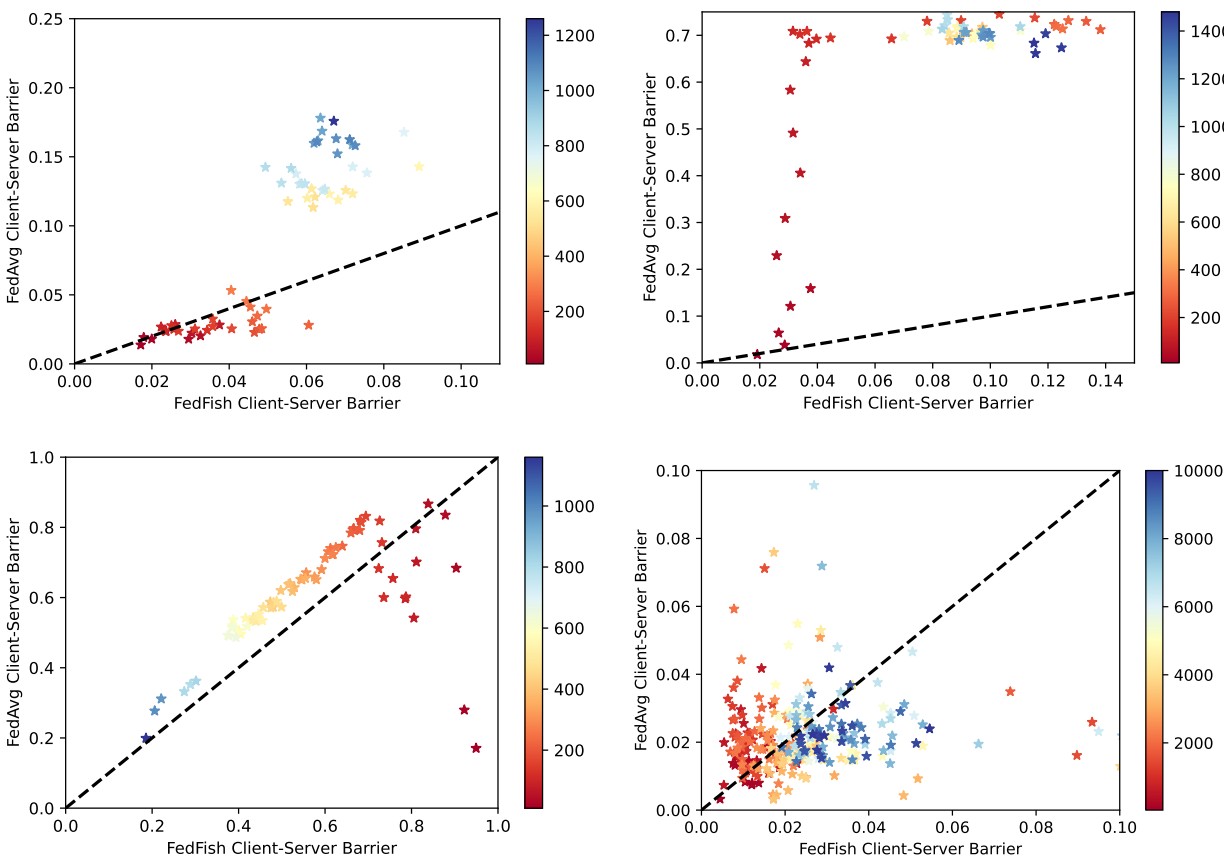

Figure 6: Measuring the Client-Server Barriers throughout training illustrates differences in how aggregating via FEDAVG or FEDFISH influences the training trajectory. Colors indicate the federated round. Stack Overflow with 8 local epochs (**top left**), Stack Overflow with 16 local epochs (**top right**), CIFAR100 with 16 local epochs (**bottom left**), and C4 with 16 local epochs (**bottom right**).

performance than FEDAVG. We observe largest gains in the case of federated pretraining on C4, followed by few-shot personalization on Stack Overflow clients, where FEDFISH improves upon FEDAVG's next-token prediction performance by 5-7%, depending on the amount of personalization data available. These results are promising since they encourage longer local training, which connotes parallelism and efficiency gains. Note, the results in fig. 5 correspond to a fixed number of federated rounds, $R$; we similarly report performance at round $R/2$ in table 5 of appendix A.4.

### 5.2.4 Client-Server Barrier

To gain more insight into the performance of FEDFISH, which only differs from FEDAVG in the aggregation step, we measure the Client-Server Barrier (CSB) defined in eq. (5), using accuracy as the metric $L$, for different model checkpoints across rounds of federated training. Note that we fix seeds to control cohort sampling in each federated round, such that the dataset iteration for FEDAVG and FEDFISH training runs match. In fig. 6, we plot this quantity for FEDFISH on the $x$-axis and for FEDAVG on the $y$-axis, using networks trained on Stack Overflow with 8 local epochs (top left), Stack Overflow with 16 local epochs (top right), CIFAR100 with 16 local epochs (bottom left) and C4 with 16 local epochs (bottom right). Points lying above or below the $x = y$ line on these plots indicate whether FEDFISH or FEDAVG, respectively, achieves lower CSB in those rounds. We observe the relative round-over-round aggregation performance of FEDAVG and FEDFISH vary significantly for different datasets and settings throughout training. For Stack Overflow, the CSB steadily increases with rounds of training, with FEDFISH achieving lower barrier than FEDAVG

during later stages of federated training. This difference is more stark when training with 16 local epochs as compared to 8 local epochs. In the case of C4, while the barrier generally increases with rounds of training, we see that FEDFISH tends to have lower values in the beginning stages of training while FEDAVG obtains lower CSB towards the final 20% of training rounds. This indicates connections to linear mode connectivity in later stages of training. We discuss this connection in appendix A.6 and leave deeper explorations to future work. Interestingly, CIFAR100 CSB values decrease with rounds of federated training with FEDFISH achieving lower barrier than FEDAVG throughout. On investigating further, we find that the client and data splits on CIFAR100 are such that each local model achieves very high performance right from the beginning of training, and maintains that performance throughout. Hence, the reduction in CSB as rounds increase is indicative of the improvement of the global model as it bridges its gap to local models. This is in contrast to the other datasets we present, wherein the local models often improve their performance more gradually.

## 5.3 Communication Cost

In general, cost of communication between clients and the server is directly related to the number of rounds of federated training as well as the number of units of (parametric) information to be exchanged. So far, we have demonstrated that FEDFISH can reduce the number of communication rounds by allowing longer local training. However, the procedure described in algorithm 1 requires clients to communicate their parameters as well as Fisher diagonals to the global server. Relative to FEDAVG, this increases communication cost of per round by a factor of two for FEDFISH. However, as demonstrated in figs. 3 and 9 (see appendix A.5), the advantage of training with FEDFISH for more local epochs can eliminate the communication overhead from our FEDFISH implementation when achieving comparable accuracy to FEDAVG in half the number of communication rounds (compare EMNIST FEDAVG with 8 local epochs after 800 communication rounds to EMNIST FEDFISH with 16 local epochs after 400 communication rounds). Alternatively, our method can be combined with adaptive federated optimization techniques (Reddi et al., 2020) so that clients only have to communicate their weighted parameters, $\text{diag}(F_i)^T \theta_i$ and the normalization step is folded into the server optimization. In this case, the communication cost of FEDFISH would be the same, per round, as that of FEDAVG. We leave this adaptive extension of our method to future work.

## 6 Related Work

**Federated Learning.** Federated Learning (FL) is a well-established paradigm, introduced in McMahan et al. (2017) and advanced through variants that account for adaptive optimization (Reddi et al., 2020), client drift (Karimireddy et al., 2020; Dandi et al., 2022) or heterogeneity (Li et al., 2020). Recent works have also explored its connections to representation learning (Collins et al., 2022) and meta-learning (Charles et al., 2023). Relevant to our function space perspective of FL are frameworks that view FL as a distributed inference problem. Al-Shedivat et al. (2021) and Guo et al. (2023) aim to approximate local posterior distributions over client parameters, deriving MCMC-based and variational inference objectives, respectively. Performance of both these methods rely on a "burn-in" period of FEDAVG training, after which the proposed algorithms are applied. The amount of burn-in training is a crucial hyperparameter and given this setting, Hou et al. (2022) find that simply chaining FEDAVG and FEDSGD is actually a theoretically sound, efficient alternative.

This work evaluates performance in settings with varied amounts of local training. At the extreme of the local training spectrum are methods that operate in the *one-shot* setting (Guha et al., 2019), to take advantage of local models that are independently trained to convergence and aggregated only once. This single-round training is a special case of FL and leads to aggregation objectives derived for optimal local models. In practice, not all clients are available simultaneously and coordinating a single round of FL is unrealistic when there are millions of clients. In contrast, the more general multi-round setting has the advantage of allowing for new clients and for clients to benefit from each other indirectly. This is the setting our work has focused on, where at each round, client models are initialized from the global model obtained in the previous round, intuitively allowing future clients to leverage information aggregated previously.

Concurrent to our work, Jhunjhunwala et al. (2023) also experiment with the one-shot setting and propose to optimize a Fisher-weighted objective to train the global model for several epochs after all clients converge

independently. In contrast, our work proposes a method for general multi-round federated training from scratch, with iterative global aggregation that is computationally equivalent to one iteration per round. The function space aggregation perspective makes no assumptions about the optimality of client models and motivates application of resulting algorithms to FL settings with multiple rounds. The resulting method is constrained to neither few local steps nor full local convergence. It is robust to local hyperparameter choices, which can otherwise be expensive and tedious to tune.

**Model aggregation.** Model aggregation has recently received attention in a number of works, most of which differ in their data and training choices during pretraining and fine-tuning, as opposed to the aggregation technique itself. For example, Wortsman et al. (2022) average parameters of models trained using different hyperparameters, random seeds, etc. Rame et al. (2023) build on this to reuse foundation models fine-tuned on an auxiliary task. In these works, fine-tuning started from the same model is likely to yield networks in the same loss basin for a new task, thus enabling parameter-space averaging and exploiting model diversity to improve performance. Model averaging has also shown up in the empirical study of Li et al. (2022) showing benefits of parallel fine-tuning of large language models on diverse data over monolithic single-model training. Similar motivations appear in Gu et al. (2023). Matena & Raffel (2022) also implement a specific Fisher-weighting, but only evaluate it in the one-shot setting to merge converged or optimal models. In fact, model averaging has been in practical use for large models at least since Vaswani et al. (2017), where final models are the result of averaging previous checkpoints. We believe our motivations for model aggregation are general and their application to these varied settings is exciting future work.

Additional literature relevant to the Client-Server Barrier evaluation criterion (section 4) is discussed in appendix A.6.

## 7 Discussion and Outlook

In this work, we provided a function-space perspective of federated learning and proposed an aggregation technique for locally trained client models based on the input-output functions they parameterize. FEDFISH is a parametric, iterative algorithm that is robust to longer local training and client data heterogeneity.

While we have highlighted the settings where FEDFISH has advantages over FEDAVG, we now discuss its limitations and possible extensions. First, FEDFISH is derived from a second-order Taylor expansion of a function space distance. There is scope to go beyond this quadratic form for better approximations (for an example, see Dhawan et al. (2023)) to derive parametric model aggregation objectives. Second, we make a diagonal approximation to the Fisher Information matrix that effectively treats each parameter as independent. It is well-understood that deep neural network parameters are highly correlated. Better approximations to the Fisher Information matrix, such as K-FAC (Martens & Grosse, 2015) or FishLeg (Garcia et al., 2023), could further boost Fisher-weighted model aggregation. However, higher dimensional approximations to the Fisher Information matrix present new challenges if applied naively, since they often increase computational burden and/or communication costs between clients and the server. Critical to compatibility with FL systems in practice, FEDFISH can also be easily combined with adaptive optimization techniques as well as differentially private training.

The demonstrated advantage of FEDFISH over FEDAVG across various large-scale settings, in terms of global performance, personalization, transfer to shifted distributions and a Client-Server Barrier metric, presents a compelling case for applying this aggregation technique more broadly. While our primary focus is FL at scale, a function-based model aggregation method and the barrier-based evaluation can be applied in any other setting where multiple models of the same architecture are trained with potentially different optimization algorithms, randomness, hyperparameters, data splits, etc. These use cases may allow for more flexibility in the use of data for merging, presenting new opportunities for improved function space aggregation.

**Acknowledgments**

The authors would like to thank Daniel M. Roy and Sewoong Oh for feedback on various drafts; Sean Augenstein for helpful discussions and Keith Rush for experiment infrastructure support.

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

# A   Appendix

## A.1   Notation

| | |
|---|---|
| $f$ | Function represented by a neural network |
| $\theta$ | Network parameters |
| $\mathcal{D}(\cdot,\cdot)$ | Function space distance |
| $\mathbf{X}$ | Input data |
| $\mathbf{y}$ | Target data |
| $\mathbf{Z}$ | Network outputs |
| $\theta_G$ | Global aggregated model |
| $N$ | Number of clients |
| $D_i$ | Dataset $(\mathbf{X}_i, \mathbf{y}_i)$ corresponding to client $i$ |
| $\theta_i$ | Locally trained model for client $i$ |
| $J_\theta^{(\mathbf{z})}$ | Output Jacobian matrix |
| $F$ | Fisher Information matrix |
| $\mathrm{diag}(F)$ | Diagonal of Fisher Information matrix |
| $L_i$ | Performance metric of interest, evaluated on data $D_i$ for client $i$ |

## A.2   FedFish Derivation

We present the complete details of our derivation for FEDFISH here.

Given $N$ client models with locally trained parameters $(\theta_i)_{i=1}^N$ and a function space distance as $\mathcal{D}(\cdot,\cdot)$ we defined the optimal global model to be:

$$\theta_G^* = \arg\min_\theta \frac{1}{N} \sum_{i=1}^N \mathcal{D}\left(f(\mathbf{X}_i;\theta), f(\mathbf{X}_i;\theta_i)\right). \tag{6}$$

We make a second-order Taylor approximation to the function space distance above with respect to model parameters $\theta$, centered at each client's $\theta_i$, the network whose outputs are to be matched. Setting $\mathcal{D}(\cdot,\cdot)$ to be the Kullback-Leibler (KL) divergence between softmax outputs of the networks, we have for each client $i$,

$$\mathcal{D}\left(f(\mathbf{X}_i;\theta), f(\mathbf{X}_i;\theta_i)\right) \approx \mathcal{D}\left(f(\mathbf{X}_i;\theta_i), f(\mathbf{X}_i;\theta_i)\right) \tag{7}$$

$$+ (\theta - \theta_i)^T \left[\left(J_\theta^{(\mathbf{z})}\right)^T \nabla_{\mathbf{Z}}\mathcal{D}\left(f(\mathbf{X}_i;\theta), f(\mathbf{X}_i;\theta_i)\right)\Big|_{\theta=\theta_i}\right] \tag{8}$$

$$+ \frac{1}{2}(\theta - \theta_i)^T \left[\nabla_\theta^2 \mathcal{D}\left(f(\mathbf{X}_i;\theta), f(\mathbf{X}_i;\theta_i)\right)\Big|_{\theta=\theta_i}\right](\theta - \theta_i), \tag{9}$$

where $J_\theta^{(\mathbf{z})}$ is the output Jacobian, arising in eq. (8) from the chain rule.

For any distance measured between the softmax outputs of the networks, $\mathcal{D}(\cdot,\cdot)$, the first term in eq. (7) is 0. Since the distance is minimized at $\theta = \theta_i$, the gradient, $\nabla_{\mathbf{Z}}\mathcal{D}\left(f(\mathbf{X}_i;\theta), f(\mathbf{X}_i;\theta_i)\right)$ in eq. (8) is also 0 when evaluated at $\theta = \theta_i$, causing this term to vanish.

Finally, applying the chain rule to the Hessian in eq. (9) yields

$$\nabla_\theta^2 \mathcal{D}(f(\mathbf{X}_i;\theta), f(\mathbf{X}_i;\theta_i))\Big|_{\theta=\theta_i} = \left(J_\theta^{(\mathbf{z})}\right)^T \nabla_\mathbf{Z}^2 \mathcal{D}(f(\mathbf{X}_i;\theta), f(\mathbf{X}_i;\theta_i)) J_\theta^{(\mathbf{z})} \tag{10}$$

$$+ \nabla_\mathbf{Z}\mathcal{D}(f(\mathbf{X}_i;\theta), f(\mathbf{X}_i;\theta_i))^T \nabla_\theta^2 f(\mathbf{X},\theta)\Big|_{\theta=\theta_i}. \tag{11}$$

Again, the second term in eq. (11) vanishes as $\nabla_\mathbf{Z}\mathcal{D}(f(\mathbf{X}_i;\theta), f(\mathbf{X}_i;\theta_i)) = 0$ when evaluated at $\theta = \theta_i$.

Since we consider models that are trained with cross-entropy loss, a natural measure of difference between outputs of two models is the KL divergence. In this case, $\nabla_\mathbf{Z}^2 \mathcal{D}_{\mathrm{KL}}(f(\mathbf{X}_i;\theta), f(\mathbf{X}_i;\theta_i))$ is the Fisher Information matrix for outputs, $F_\mathbf{Z}$. Via chain rule, the first term above $\left(J_\theta^{(\mathbf{z})}\right)^T F_\mathbf{Z} J_\theta^{(\mathbf{z})} = F_\theta$ is simply the Fisher Information matrix for the network parameters. Henceforth, we simply denote this as $F$ or $F_i$ to indicate the Fisher Information matrix corresponding to parameters $\theta_i$ of client $i$.

Our final approximation to function space distance reduces to

$$\mathcal{D}(f(\mathbf{X}_i;\theta), f(\mathbf{X}_i;\theta_i)) \approx \frac{1}{2}(\theta - \theta_i)^T F_i (\theta - \theta_i) \tag{12}$$

$$\approx \frac{1}{2}\sum_{j=1}^{|\theta_i|} F_i^{(j)}(\theta^{(j)} - \theta_i^{(j)})^2. \tag{13}$$

Here, eq. (13) makes a diagonal approximation to the Fisher Information matrix, with $F_i^{(j)}$ denoting the $j$-th diagonal entry of $F_i$.

Plugging eq. (13) into the optimization problem of eq. (6) gives

$$\theta_G^* = \arg\min_\theta \frac{1}{2N}\sum_{i=1}^{N}\sum_{j=1}^{|\theta_i|} F_i^{(j)}(\theta^{(j)} - \theta_i^{(j)})^2. \tag{14}$$

Equation (4) is now a convex optimization problem, which we can solve by taking its gradient with respect to $\theta$ and setting it to 0. It has the following closed-form solution:

$$\theta_G^* = \frac{\sum_{i=1}^{N}\mathrm{diag}(F_i)^T \theta_i}{\sum_{i=1}^{N}\mathrm{diag}(F_i)}, \tag{15}$$

### A.3 Experimental Details

#### A.3.1 Datasets, Tasks & Models

We use four datasets for training models using FEDAVG and FEDFISH: the federated extended MNIST dataset (EMNIST) (Cohen et al., 2017), the CIFAR100 dataset (Krizhevsky et al.), the Stack Overflow dataset (Authors, 2019) and the C4 dataset (Raffel et al., 2020). We additionally use the CC-News (Hamborg et al., 2017) and Stack Overflow datasets for evaluating transfer and post-personalization performance of models trained on C4 using each algorithm of study.

Each of these datasets is publicly available. EMNIST is licensed under Standard Reference Data by NIST. CIFAR100 is published by the authors. Stack Overflow is licensed under the Creative Commons Attribution-ShareAlike 3.0 Unported License. C4 and CC-News are hosted by `commoncrawl.org` and we access both through HuggingFace datasets.

Table 1 lists the scale of each dataset, the associated task and the model used for training. We include additional details on dataset preprocessing and model configuration for each experiment setting below.

Table 1: Datasets, Tasks & Models

| Dataset | Num Clients | | Num Examples | | Task | Model |
|---|---|---|---|---|---|---|
| | Train | Test | Train | Test | | |
| EMNIST | 3.4K | 3.4K | 672K | 77K | Character Recognition | CNN |
| CIFAR100 | 500 | 100 | 50K | 10K | Image Recognition | ResNet-18 with GroupNorm |
| Stack Overflow | 342K | 204K | 135.8M | 16.6M | Next-Token Prediction | 350M Parameter Decoder-only Transformer |
| C4 | 15.6M | 8.5K | 364.9M | 365K | Next-Token Prediction | 1.5B Parameter Decoder-only Transformer |
| CC-News | – | 8.8K | – | 708K | Next-Token Prediction | 1.5B Parameter Decoder-only Transformer |

**EMNIST** The EMNIST dataset is comprised of 28x28 grey-scale pixel images of alphanumeric handwritten characters. There are 62 characters represented. The dataset has natural heterogeneity stemming from characters being written by different authors. We partition the handwritten characters in EMNIST according to their author, as proposed by Caldas et al. (2018). We train a two-layer LeNet CNN model (Lecun et al., 1998) for character recognition: two convolutional layers with 3x3 kernels and strides of length 1, a max pooling layer using dropout with $p = 0.25$, a dense layer with 128 units and dropout with $p = 0.5$, and a final dense output layer.

**CIFAR100** The CIFAR100 dataset consists of 32x32x3 pixel images with one of 100 labels. We preprocess the images using standard data augmentations, including padding to 36x36 dimensions, randomly cropping to 32x32, randomly flipping along the vertical axis and applying normalization. We partition CIFAR100 according to the two-level Dirichlet allocation scheme proposed by Reddi et al. (2020). We train a standard ResNet-18 model with the batch normalization layers replaced with group normalization layers, following Reddi et al. (2020).

**Stack Overflow** The Stack Overflow dataset is a language-modeling dataset consisting of question–answer pairs from `stackoverflow.com`. Each client corresponds to a user on the platform. The data is split into train, test and validation: train client examples are from before 2018-01-01 UTC, test client examples are from after 2018-01-01 UTC, and validation clients are held out from both train and test splits. We train a 350M parameter decoder-only transformer model on the train split of Stack Overflow. We use the validation client split in our evaluations of both the Stack Overflow base model and the C4 base model (for assessing transfer performance, see below).

**C4** The Colossal Clean Crawled Corpus (C4) dataset is a cleaned version of Common Crawl's web crawl corpus (Raffel et al., 2020). We use the federated version of this dataset presented in Charles et al. (2023),

| Dataset | EMNIST | CIFAR100 | Stack Overflow | C4 |
|---|---|---|---|---|
| **Number of clients per round** | 64 | 64 | 16 | 8 |
| **Number of training rounds** | 1500 | 1500 | 1500 | 10000 |
| **Batch size for local training** | 10 | 25 | 4 | 4 |
| **Max client dataset size per round** | 100 | 100 | 16 | 16 |
| **Sequence length (training)** | - | - | 128 | 1024 |
| **Sequence length (personalization and evaluation)** | - | - | 128 | 128 |
| **Number of personalization epochs** | 1 | 4 | 4 | 4 |
| **Global learning rate** | 1e-3 | 1e-3 | 1e-2 | 1e-2 |
| **Local learning rate** | 1e-3 | 1e-3 | 1e-3 | 1e-3 |

Table 2: Final hyperparameter configurations for all datasets.

where each client corresponds to a different domain name (i.e., `nytimes.com`. We train a 1.5B parameter decoder-only transformer model on the train split of federated C4.

We evaluate the C4 base model on a federated version of the C4 test split, as well as federated CC-News and Stack Overflow to assess transfer performance. CC-News is similarly split by domain name, using Dataset Grouper (Charles et al., 2023). We further measure few-shot performance of the C4 base model by conducting a personalization evaluation on 25% of held-out client data for each dataset of interest (C4, CC-News, Stack Overflow). Because of this specific personalization evaluation, we filter C4 evaluation datasets to have at least 4 examples per client.

### A.3.2 Federated Algorithm Configuration and Hyperparameters

We implement the federated algorithms, FedAvg and FedFish, such that a global model is broadcast to a number or clients at each round, client train their models locally, model deltas are returned as pseudo-gradients to the global server for aggregation, and a global optimizer is used to make a single update on the global model using the aggregated pseudo-gradients as its own gradient.

**Optimizers.** We follow standard configurations used in Charles et al. (2023), with stochastic gradient descent as the local optimizer and Adam as the global server optimizer, across experiments.

**Hyperparameter Tuning** We fix hyperparameters like number of clients per round, number of training rounds, local batch size, maximum dataset size for any client and sequence length for language models to reasonable values based on previous literature. For local and global learning rates, we conducted a grid search over [1e-4, 5e-4, 1e-3, 5e-3, 1e-2, 5e-2, 1e-1], and chose the best performing hyperparameters. Final hyperparameter configurations used to obtain the results in the paper are listed in table 2. These are held consistent for FedAvg and FedFish experiments.

### A.3.3 Hardware Configuration

We run image-classification experiments on a TPU Pod slice consisting of 4 TPU v2 chips in a 2x2 topology, interconnected on a single machine. Each TPU v2 chip contains two TensorCores, 16 GiB of high-bandwidth memory, and 90.75 GiB RAM. We run language-modeling experiments on a TPU Pod slice consisting of 16 TPU v3 chips in a 4x4 topology, configured to use a multi-machine inter-chip interconnect mesh. Each TPU v3 chip contains two TensorCores, 32 GiB of high-bandwidth memory, 87.25 GiB RAM, 900 GBps bandwidth, and 123 teraflops peak compute.

| Dataset | Local epochs | Method | Global Accuracy | | Personalized Accuracy | |
|---|---|---|---|---|---|---|
| | | | Fixed compute | Fixed rounds | Fixed compute | Fixed rounds |
| EMNIST | 4 | FedAvg | 81.38 | 81.38 | 83.09 | 83.09 |
| | | FedFish | 82.64 | 82.64 | 83.44 | 83.44 |
| | 8 | FedAvg | 82.05 | 82.24 | 83.46 | 83.82 |
| | | FedFish | 83.19 | 84.42 | 84.37 | 85.6 |
| | 16 | FedAvg | 76.45 | 77.52 | 77.96 | 79.63 |
| | | FedFish | 79.01 | 82.86 | 81.05 | 84.96 |
| CIFAR100 | 4 | FedAvg | 35.75 | 35.75 | 37.06 | 37.06 |
| | | FedFish | 36.88 | 36.88 | 41.31 | 41.31 |
| | 16 | FedAvg | 28.56 | 33.28 | 36.00 | 32.13 |
| | | FedFish | 31.63 | 37.72 | 40.53 | 44.69 |

Table 3: Longer local training improves overall performance in terms of both global model accuracy as well as personalization. When amount of compute, i.e. total number of training epochs, is fixed, FEDFISH can achieve better performance than FEDAVG with fewer rounds of communication between the server and clients.

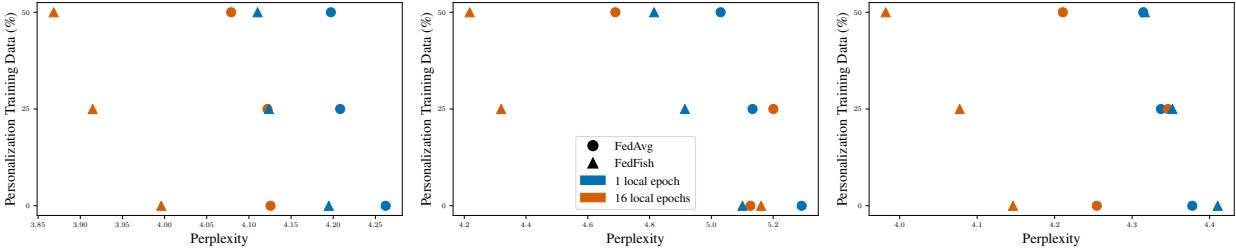

Figure 7: Transfer performance on perplexity (lower is better) after federated pretraining on C4 and evaluating on C4 (**left**), Stack Overflow (**center**) and CC-News (**right**).

## A.4 Additional Results

Here, we include additional results and report the performance visualized in section 5. Table 3 shows results on image benchmarks where performance is computed by keeping either the computational requirement or the total number of rounds fixed. For the former, we limit the number of training rounds according to the number of local epochs so that total amount of client-side training remains constant. Unsurprisingly, allowing for more rounds of training helps performance. However, at fixed compute, we see that FEDFISH is able to achieve better performance than FEDAVG with fewer rounds of communication. Table 4 and table 5 include full results on our language modeling experiments, including different numbers of local training epochs, different datasets used for federated pretraining or personalization, different performance metrics and different amounts of data used for personalization. Finally, similar to fig. 5 with shows next-token prediction performance, we present perplexity performance for the C4 transfer experiments in appendix A.3.3.

## A.5 Efficiency of FedFish

We recognize that there are opportunities to improve the efficiency of FEDFISH, both in terms of computation and communication. As presented in algorithm 2, each client takes an additional pass over its data to estimate

| Training Dataset | Training Local Epochs | Personalization Dataset | Method | Global | | Personalized (25%) | | Personalized (50%) | |
|---|---|---|---|---|---|---|---|---|---|
| | | | | Token Pred | Perplexity | Token Pred | Perplexity | Token Pred | Perplexity |
| Stack Overflow | 8 | Stack Overflow | FedAvg | 32.00 | 3.79 | - | - | 32.34 | 3.90 |
| | | | FedFish | 32.98 | 3.77 | - | - | 34.28 | 3.68 |
| | 16 | | FedAvg | 21.77 | 5.56 | - | - | 25.46 | 4.61 |
| | | | FedFish | 31.27 | 3.96 | - | - | 32.60 | 3.81 |
| C4 | 1 | C4 | FedAvg | 30.16 | 4.26 | 30.81 | 4.21 | 30.95 | 4.20 |
| | | | FedFish | 31.15 | 4.19 | 32.10 | 4.12 | 32.34 | 4.11 |
| | 16 | | FedAvg | 31.04 | 4.13 | 32.26 | 4.12 | 32.73 | 4.08 |
| | | | FedFish | 32.37 | 3.99 | 33.54 | 3.92 | 33.98 | 3.87 |
| C4 | 1 | Stack Overflow | FedAvg | 17.91 | 5.29 | 19.85 | 5.13 | 21.58 | 5.03 |
| | | | FedFish | 19.55 | 5.101 | 22.07 | 4.91 | 23.38 | 4.815 |
| | 16 | | FedAvg | 18.83 | 5.13 | 19.63 | 5.201 | 22.98 | 4.69 |
| | | | FedFish | 19.54 | 5.16 | 26.89 | 4.32 | 27.77 | 4.22 |
| C4 | 1 | CC-News | FedAvg | 29.03 | 4.38 | 29.45 | 4.34 | 29.73 | 4.31 |
| | | | FedFish | 29.53 | 4.41 | 30.14 | 4.35 | 30.61 | 4.32 |
| | 16 | | FedAvg | 29.96 | 4.25 | 29.69 | 4.35 | 31.00 | 4.21 |
| | | | FedFish | 31.00 | 4.15 | 31.75 | 4.08 | 32.58 | 3.98 |

Table 4: Full results on global and post-personalization performance of language models that are pretrained in a federated manner with varying number of local epochs and then evaluated on different datasets. These results correspond to the entire $R$ rounds of federated training.

| Training Dataset | Training Local Epochs | Personalization Dataset | Method | Global | | Personalized (25%) | | Personalized (50%) | |
|---|---|---|---|---|---|---|---|---|---|
| | | | | Token Pred | Perplexity | Token Pred | Perplexity | Token Pred | Perplexity |
| C4 | 1 | C4 | FedAvg | 26.82 | 4.67 | 27.56 | 4.61 | 27.73 | 4.60 |
| | | | FedFish | 27.69 | 4.63 | 28.70 | 4.55 | 28.91 | 4.54 |
| | 16 | | FedAvg | 28.63 | 4.42 | 29.84 | 4.37 | 30.11 | 4.39 |
| | | | FedFish | 30.33 | 4.28 | 31.54 | 4.11 | 31.61 | 4.10 |
| | 1 | Stack Overflow | FedAvg | 17.10 | 5.38 | 19.92 | 5.20 | 21.78 | 5.10 |
| | | | FedFish | 17.25 | 5.41 | 19.62 | 5.2 | 21.10 | 5.08 |
| | 16 | | FedAvg | 18.57 | 5.29 | 16.23 | 7.03 | 18.91 | 6.09 |
| | | | FedFish | 17.80 | 5.42 | 20.81 | 4.77 | 22.78 | 4.58 |
| | 1 | CC-News | FedAvg | 26.03 | 4.79 | 26.44 | 4.75 | 26.72 | 4.72 |
| | | | FedFish | 26.23 | 4.86 | 26.88 | 4.80 | 27.34 | 4.76 |
| | 16 | | FedAvg | 27.55 | 4.58 | 27.23 | 4.59 | 28.02 | 4.55 |
| | | | FedFish | 28.91 | 4.44 | 29.86 | 4.29 | 30.06 | 4.27 |

Table 5: Results on global and post-personalization performance of language models in different settings, evaluated at $R/2$ rounds of federated training.

the Fisher diagonal (lines 9-12) and communicates this estimate along with its model update (line 13). This results in an increase in computational cost equivalent to one epoch of training and a two-fold increase in communication cost.

### A.5.1 Computational Overhead of FedFish

To eliminate the need for an additional forward and backward pass through the model to compute Fisher information, we can instead use the gradients from the last local epoch of training. This is a further approximation as the gradients are with respect to the evolving client model rather than with respect to

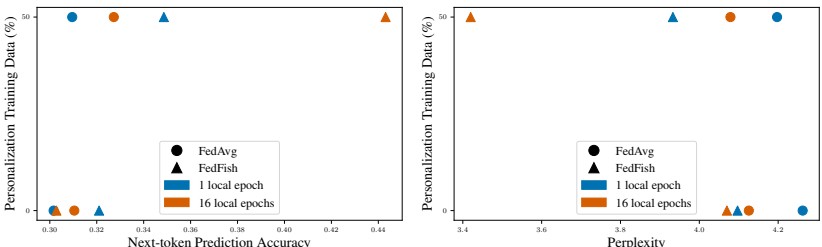

Figure 8: Global and post-personalization performance in terms of next-token prediction (**left**) and perplexity (lower is better) (**right**) after federated pretraining on C4 using FEDAVG or FEDFISH where the Fisher is estimated using gradients from epoch $E$.

| Training Dataset | Training Local Epochs | Personalization Dataset | Method | Global | | Personalized (50%) | |
|---|---|---|---|---|---|---|---|
| | | | | Token Pred | Perplexity | Token Pred | Perplexity |
| C4 | 1 | C4 | FedAvg | 30.16 | 4.26 | 30.95 | 4.20 |
| | | | FedFish | 32.10 | 4.10 | 34.86 | 3.93 |
| | 16 | | FedAvg | 31.04 | 4.13 | 32.72 | 4.08 |
| | | | FedFish | 30.28 | 4.07 | 44.31 | 3.42 |

Table 6: Results from ablation study of FEDFISH without computational overhead, where the Fisher is estimated using gradients from epoch $E$. Global and post-personalization performance of model pretrained and personalized on C4 evaluated at $R$ rounds of federated training.

the fully updated client model that will be merged. The computational overhead of FEDFISH, as presented in algorithm 2, is likely to be more of a hindrance as models scale. Given this efficiency concern is most relevant for the largest setting we consider, we perform an ablation study on C4 to investigate the effect of making the approximation described above.

We run FEDFISH C4 experiments using gradients from the last epoch of local training in each round to compute the Fisher diagonal. Results are presented in fig. 8 and table 6. We find that this further approximation does not reduce performance - even when only training with a single pass over the data. Surprisingly, we see a substantial improvement in personalization performance for training with 16 local epochs when using gradients from the 16[th] epoch rather than from an additional pass after fixing the local model.

### A.5.2 Communication Overhead of FedFish

Note that overall communication cost is not just measured in bits communicated per round, but also in the total number of communication rounds required. Depending on the network constraints one of these factors may be more critical to minimize. As discussed in section 5.3, the presented implementation of FEDFISH (see algorithm 1 and algorithm 2) has twice the communication cost (per federated round) of FEDAVG. This overhead can be eliminated by combining our method with adaptive federated optimization techniques (Reddi et al., 2020), having clients only send their weighted parameters and folding the normalization into the server optimization. We leave this extension to future work, and provide a more critical look at the advantage of FEDFISH given the presented implementation.

**Ablation: Communication Rounds** Figure 9 depicts the evaluation performance of FEDAVG and FED-FISH on EMNIST across local epochs normalized by the number of communication rounds. FEDFISH converges faster and to a higher global accuracy and post-personalization accuracy than FEDAVG. Despite the additional cost per round of FEDFISH, convergence over fewer communication rounds is useful for a setting in which the network is not bandwidth-constrained but clients are intermittently reachable.

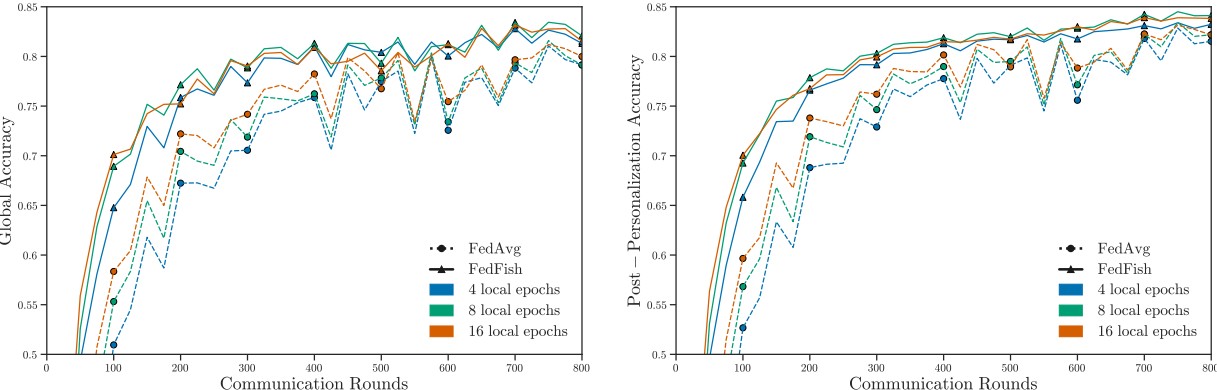

Figure 9: When normalizing by communication rounds, training on EMNIST with FEDFISH converges faster and to a higher global accuracy (**left**) and post-personalization accuracy (**right**) than with FEDAVG. Results are shown for experiments across varying numbers of local epochs. Each marker indicates 100 federated communication rounds.

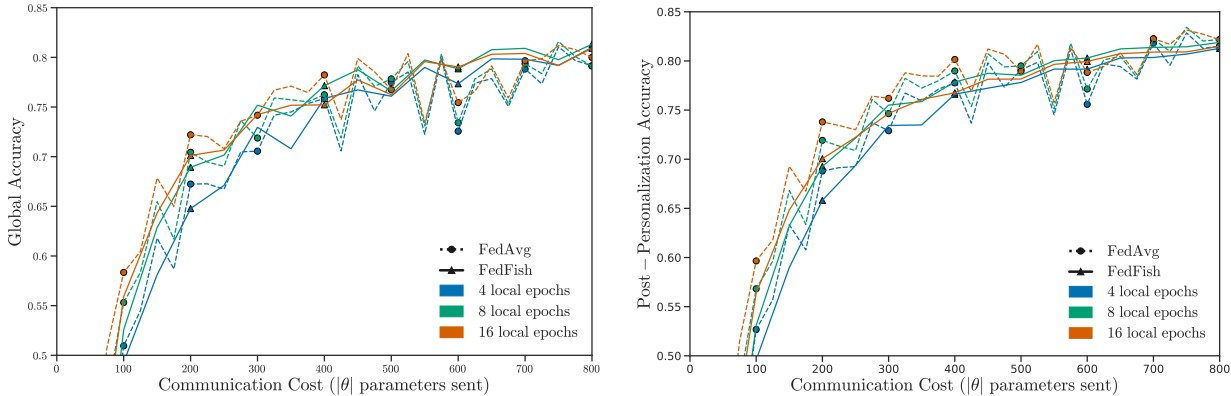

Figure 10: When normalizing by communication cost, training on EMNIST with FEDFISH matches the performance of FEDAVG both in terms of global accuracy (**left**) and post-personalization accuracy (**right**), with half the number of communication rounds. Results are shown for experiments across varying numbers of local epochs. Each marker indicates 100 federated communication rounds.

**Ablation: Communication Cost** Figure 10 depicts the evaluation performance of FEDAVG and FED-FISH on EMNIST across local epochs normalized by the total communication cost (measured in units of $|\theta|$). FEDFISH performs comparably to FEDAVG both in global accuracy and post-personalization accuracy. Taking the additional cost per round of FEDFISH into account, FEDFISH is no more costly than FEDAVG measured in terms of bits communicated to accuracy achieved.

### A.6 Additional Related Works

**Linear Mode Connectivity.** Frankle et al. (2020) The Client-Server Barrier is largely inspired by the error barrier definition in Fort et al. (2020). There the authors define an error barrier as the maximum increase in error on a linear path in the parameter space between two models. There are a few key differences between the error barrier and client-server barrier: they consider models trained on the same training data; using our notation, the losses $L_i$ are identical for all $i$; $i$ indexes over different models $\theta_i$ (e.g., models trained in the same centralized way but independently). The error barrier corresponds then to the maximum over

$\{w_i\}_{i=1,\ldots,N}$, where $w_i$ is the weight corresponding to $\theta_i$ when linearly interpolating between $\{\theta_i\}_{i=1,\ldots,N}$. Prior to Fort et al. (2020), error barriers appeared under "instability" definition in Frankle et al. (2020) for evaluating how different trajectories of a pair of networks connect in the loss landscape. Later, similar metrics were used in other model averaging work, such as Wortsman et al. (2022), where the authors consider multiple networks trained with different hyperparameters.

