# OpenReview forum: "Leveraging Function Space Aggregation for Federated Learning at Scale"
_TMLR — Accepted by TMLR_

### Review · Reviewer_uX9A · 2023-12-03

**Summary Of Contributions:**

The paper provides a function-space perspective of federated learning and proposes the FedFish algorithm that uses fisher-weighted averaging for aggregation of the local models. In particular, it replaces the weighted averaging set of FedAvg with fisher-weighted averaging. FedFish is robust to longer local training and client data heterogeneity. Authors demonstrate the advantage of FedFish over FedAvg across various large-scale settings, in terms of global performance, personalization, and transfer to shifted distributions, and a Client-Server Barrier metric presents a compelling case for applying this aggregation technique more broadly.

**Audience:**

Yes

**Claims And Evidence:**

Yes

**Requested Changes:**

1. Plotting Fig. 3 results with respect to communication bits would help in a better understanding of iso-communication comparison.
2. Comparison with Federated Learning baselines designed for heterogeneous data such as FedProx [1], ScaffNew[2], etc can strengthen the argument that FedFish is more robust to client data heterogeneity.

[1] Li, Tian, et al. "Federated optimization in heterogeneous networks." Proceedings of Machine learning and systems 2 (2020): 429-450.

[2] Mishchenko, Konstantin, et al. "Proxskip: Yes! local gradient steps provably lead to communication acceleration! finally!." International Conference on Machine Learning. PMLR, 2022.

**Strengths And Weaknesses:**

Strengths:
1. Proposed a new federated learning algorithm referred to as FedFish.
2. Formalizes a function space perspective of federated learning to motivate FedFish.
3. FedFish is robust to longer local training and client data heterogeneity.
4. In the extreme case of completely disjoint supports, FedFish matches the locally learned functions of the clients on their respective
input data. In this case, FedFish performs significantly better than FedAvg.
5. FedFish results in global networks that are more amenable to efficient personalization via local fine-tuning on the same or shifted data distributions.
6. Experimental results across several settings in image and language benchmarks are presented.
7. The paper is well-written and easy to follow.

Weaknesses/Questions:
1. FedFish is a trivial extension of fisher-weighted averaging proposed in [1] to federated learning setups.
2. FedFish requires $2\times$ communication (from client to server) as compared to FedAvg in each communication round.
3. Needs an additional forward and backward pass through the model to compute fisher information. Can we just use the last local iteration's gradients (Epoch E's gradient in Algorithm 2) to compute fisher information?
4. Theoretical analysis for the convergence rate of FedFish is missing.
5. In Fig. 3a, the curve for FedAvg with 4 local epochs overlaps with FedFish with 8 local epochs and the curve for FedAvg with 8 local epochs overlaps with FedFish with 16 local epochs. Is it possible that the FedFish converges at the same rate as FedAvg for iso-communication comparison?
6. The squared gradient is also used as a proxy for sharpness estimate. It would be interesting to correlate FedFish's aggregation mechanism to the sharpness or flatness of an individual client's loss surface. Is this aggregation somehow helping us to achieve a flatter loss region for the global model and hence allowing for longer local training?

[1] Matena, Michael S., and Colin A. Raffel. "Merging models with fisher-weighted averaging." Advances in Neural Information Processing Systems 35 (2022): 17703-17716.

---

> ### Author Response · Authors · 2023-12-27
> **Response to Reviewer uX9A (Part 1)**
>
> We thank the reviewer for their concise summary of our work and constructive feedback on the paper! We are also grateful that they found our claims and empirical analysis to be substantiated and relevant to the community. Below, we address the questions and concerns raised, along with requested changes to the manuscript.
>
> > FedFish is a trivial extension of fisher-weighted averaging proposed in [1] to federated learning setups.
>
> While it is true that FedFish and Fisher-weighted averaging in [1] have very similar forms for the aggregation step, we would like to highlight a few differences between the two works. 1). As the reviewer pointed out, our work considers realistic federated learning scenarios. In particular, we aim to evaluate function space aggregation for federated learning where the individual models are trained on heterogeneous data distributions and different clients are available during different rounds. As opposed to this, [1] considers models that are trained to convergence on the same data distribution with different hyperparameters and are all simultaneously available for aggregation.  2). We demonstrate, using the function space perspective, that FedFish can be used to aggregate non-converged parameters in multiple rounds, and further empirically validate this claim across various settings. In contrast, the analysis of [1] relies on the assumption that the models to be merged are optimal for their training data. Accordingly, they only consider the “one-shot” setting where all models must be trained to convergence and aggregated once at the very end. 3). Finally, there is a subtle difference in the implementations of the two methods, specifically with respect to dealing with numerical instability when the denominator in the aggregation step formula is close to zero. According to their paper, [1] “choose a privileged target model in all of our experiments and default to the parameter’s value in the target model in these cases”. In the case of FedFish, we instead use the computed Fisher coefficient for all parameters and add a small value to each to avoid the numerical instability. Since the same value is added to each Fisher coefficient, we believe this does not significantly change the relative importance of parameters.
>
> > FedFish requires 2× communication (from client to server) as compared to FedAvg in each communication round.
>
> We agree with the reviewer and point out this limitation of FedFish in our work. Note that the increase in communication cost results from the normalization term, where the Fisher-weighted average of model parameters is divided by the sum of all Fisher coefficients. Since this denominator is independent of the individual client parameters (it sums over them), we propose that the normalization term may be folded into the server learning rate used during aggregation. Adaptive optimization techniques may be used to account for this normalization term, such that only Fisher-weighted parameters are required to be communicated in each round, incurring the same cost as FedAvg.
>
> > Needs an additional forward and backward pass through the model to compute fisher information. Can we just use the last local iteration's gradients (Epoch E's gradient in Algorithm 2) to compute fisher information?
>
> Thank you to the reviewer for this suggestion! The last local epoch’s gradients would indeed be a reasonable and computationally cheaper way to implement FedFish. We evaluated the performance of this variant for the C4 dataset, plotted in Figure 8 of the updated manuscript. We find no significant decrease in the global next-token prediction accuracy and in fact surprisingly, observe an improvement after personalization.
>
> Requested Changes:
>
> > Plotting Fig. 3 results with respect to communication bits would help in a better understanding of iso-communication comparison.
>
> We thank the reviewer for this suggestion since a performance comparison with respect to communication bits would provide insights into the benefits of FedFish. We visualized the requested plot and added it to the updated paper as Figure 10. We find that when normalizing by communication cost, training on EMNIST with FedFish matches the performance of FedAvg both in terms of global accuracy (left) and post-personalization accuracy (right), with half the number of communication rounds.

---

> > ### Author Response · Authors · 2023-12-27
> > **Response to Reviewer uX9A (Part 2)**
> >
> > Requested Changes (cont.):
> >
> > > Comparison with Federated Learning baselines designed for heterogeneous data such as FedProx [1], ScaffNew[2], etc can strengthen the argument that FedFish is more robust to client data heterogeneity.
> >
> > We agree with the reviewer that including comparisons to other methods aimed at addressing client data heterogeneity would strengthen the work. Given that the aim of our work is to strictly examine the aggregation strategy of federated algorithms, to this end, we think that FedAvg is a reasonable baseline and point of comparison. Various works ([1], [2]) have shown that while FedProx [3] is a bit easier to tune, it does not lead to any fundamental improvements over FedAvg. ScaffNew [4] does not directly support cross-device settings with partial participation, as it requires every client to participate in every round of training, which is not compatible with our setting.
> >
> > Because our proposed strategy for mitigating the challenges of client heterogeneity involves simply changing the aggregation weighting, an additional advantage of FedFish is that it can be composed with other approaches that rely on parameter averaging. Fisher-weighting can be used as an alternative to simple parameter averaging in any setting.
> >
> > Again, we thank the reviewer for their time and effort to summarize our contributions and make valuable suggestions that strengthen our work. Please let us know if there are any additional updates or clarifications we can provide.
> >
> > [1] Charles, Zachary, and Jakub Konečný. "Convergence and accuracy trade-offs in federated learning and meta-learning." International Conference on Artificial Intelligence and Statistics. PMLR, 2021.
> >
> > [2] Wang, Jianyu, et al. "Tackling the objective inconsistency problem in heterogeneous federated optimization." Advances in neural information processing systems 33 (2020): 7611-7623.
> >
> > [3] Li, Tian, et al. "Federated optimization in heterogeneous networks." Proceedings of Machine learning and systems 2 (2020): 429-450.
> >
> > [4] Mishchenko, Konstantin, et al. "Proxskip: Yes! local gradient steps provably lead to communication acceleration! finally!." International Conference on Machine Learning. PMLR, 2022.

---

> > > ### Author Response · Authors · 2023-12-27
> > > **Note on sharpness in response to Reviewer uX9A**
> > >
> > > The reviewer brought up a very interesting question about the relationship between federated aggregation and sharpness of client loss landscapes.
> > >
> > > > The squared gradient is also used as a proxy for sharpness estimate. It would be interesting to correlate FedFish's aggregation mechanism to the sharpness or flatness of an individual client's loss surface. Is this aggregation somehow helping us to achieve a flatter loss region for the global model and hence allowing for longer local training?
> > >
> > > Sharpness is certainly an interesting property to track and may provide some insight into the effect of each aggregation method on the local loss landscapes. As the reviewer suggested, we tracked the squared gradient of the broadcast global model over the course of training with respect to each client’s data, averaging across clients sampled in that round. We performed this experiment on EMNIST, trained with both FedAvg and FedFish, using 4 or 16 local epochs. We note that the empirical squared gradient is a fairly crude approximation of sharpness, and so there is further work needed to validate these findings. Nevertheless, our preliminary results show that FedFish aggregation yields a global model with higher sum of squared gradients across clients than FedAvg aggregation; and longer local training similarly results in higher sum of squared gradients than using fewer local epochs for both algorithms. If this measure reflects the sharpness of loss landscapes well, we hypothesize that increased sharpness resulting from FedFish and longer local training may explain the improved personalization compared to FedAvg. If FedFish lands in areas of higher curvature in the client-specific loss landscapes, it may be able to make more progress when clients take gradient steps during personalization. We provide initial results in the table below, but note that a more in-depth investigation would be necessary to make conclusions about the optimization landscape.
> > >
> > > | Federated Round | Method  | Local Training Epochs | Sum of Squared Gradients |
> > > | --------------- | ------- | --------------------- | ------------------------ |
> > > | 1               | FedAvg  | 4                     | 11.68                    |
> > > | 1               | FedAvg  | 16                    | 11.82                    |
> > > | 1               | FedFish | 4                     | 10.55                    |
> > > | 1               | FedFish | 16                    | 10.37                    |
> > > | 130             | FedAvg  | 4                     | 181.06                   |
> > > | 130             | FedAvg  | 16                    | 187.51                   |
> > > | 130             | FedFish | 4                     | 490.33                   |
> > > | 130             | FedFish | 16                    | 420.53                   |
> > > | 306             | FedAvg  | 4                     | 280.58                   |
> > > | 306             | FedAvg  | 16                    | 300.05                   |
> > > | 306             | FedFish | 4                     | 679.87                   |
> > > | 306             | FedFish | 16                    | 669.11                   |
> > > | 481             | FedAvg  | 4                     | 364.20                   |
> > > | 481             | FedAvg  | 16                    | 409.56                   |
> > > | 481             | FedFish | 4                     | 865.65                   |
> > > | 481             | FedFish | 16                    | 960.14                   |
> > > | 706             | FedAvg  | 4                     | 440.92                   |
> > > | 706             | FedAvg  | 16                    | 539.34                   |
> > > | 706             | FedFish | 4                     | 1,080.04                 |
> > > | 706             | FedFish | 16                    | 1,239.33                 |
> > > | 961             | FedAvg  | 4                     | 497.10                   |
> > > | 961             | FedAvg  | 16                    | 666.59                   |
> > > | 961             | FedFish | 4                     | 1,220.29                 |
> > > | 961             | FedFish | 16                    | 1,491.81                 |

---

### Review · Reviewer_joh1 · 2023-12-13

**Summary Of Contributions:**

This paper proposes a novel federated learning algorithm based on the function space. Instead of taking simple weighted average in the traditional FedAvg method, the proposed algorithm, FedFish,  aggregates local approximations to the functions learned by clients, using an estimate based on their Fisher information. To validate their scheme, the authors utilize several models and datasets for the detailed performance comparison. The results show that FedFish is more robust to longer local training. Also, FedFish benefits the efficient personalization via local fine-tuning on the same or shifted data distributions.

**Audience:**

Yes

**Claims And Evidence:**

Yes

**Requested Changes:**

1. Please add theoretical analysis for the FedFish.
2. Please add technical discussion for the empirical results to explain why FedFish outperforms FedAvg.
3. It would be great to see the results on a larger image dataset, such as ImageNet.

**Strengths And Weaknesses:**

Strengths:
1. The investigated topic is very interesting based on the function space aggregation.
2. The paper is well written and easy to follow. The presentation is clear.
3. The empirical studies are extensive and comprehensive.

Weaknesses:
1. The idea of function space aggregation using Fisher information matrix is existing. Please see [1]. Hence, the novelty in this work seems low. Though in this work the setting is federated learning. The optimal solution $\theta^*$ is almost exactly the same.
2. The theoretical contributions in this paper are not strong. If the authors could show some theoretical analysis of how FedFish enables better robustness, that would make the paper more technically solid and sound.
3. From the experimental results, we know that FedFish performs better if the local training is longer. But the authors only presented the results, failing to discuss why this was the case. Also, for other results, the authors only described the outcomes without clearly discussing the reasons.

---

> ### Author Response · Authors · 2023-12-27
> **Response to Reviewer joh1**
>
> Thank you to the reviewer for their summary or our paper and valuable comments, each of which are addressed below.
>
> > Please add theoretical analysis for the FedFish.
>
> While the main contributions of this work are empirical, the function space perspective is motivated by its benefits over parameter space aggregation and the particular instantiation of our algorithm is derived in Appendix A.2. Specifically, function space aggregation takes into account the data distributions that client models are trained on, resulting in global models that better capture the functions learned by them.
>
> > Please add technical discussion for the empirical results to explain why FedFish outperforms FedAvg.
>
> The FedFish algorithm and the derivation of its objective are motivated by using function space aggregation as an alternative to parameter space aggregation. Performance of FedAvg has been found to suffer with longer local training due to the “client drift” phenomenon, due to which a simple average of client parameters no longer captures the information learned by each client during its local training. We derived FedFish as an approximation to a global model which matches the functions learned by each client on its respective training distribution, and hypothesized that this would mitigate client drift better than FedAvg. We measured and plotted the Client-Server Barrier throughout training to validate our hypothesis that FedFish learns global models that better match the functions learned by clients. Our experiments, including downstream personalization and transfer performance, further investigate the effect of such pretraining in different settings.
>
> > It would be great to see the results on a larger image dataset, such as ImageNet.
>
> We agree that it would be useful to evaluate performance at scale on image data, in addition to the large scale language data (C4) experiments we provide. However, there is currently a lack of large scale federated image datasets readily available. We excluded ImageNet from our image datasets since it does not have any inherent federated structure. There is a need to first define a partitioning of ImageNet that represents a realistic federated setting. Constructing and evaluating this setting would take more time and resources than we had available during the discussion period.
>
> We believe federated C4 to be the largest dataset scale currently available in the federated learning literature and representative of a realistic setting [1]. We hope that our set of experiments ranging from toy to large scale (C4), and spanning the language and image domains, is sufficiently comprehensive given existing federated datasets.
>
> We thank the reviewer again for their feedback on our paper and are grateful it was found interesting and validated with extensive experimentation. We welcome discussion and would be happy to provide any further clarifications.
>
> [1] Zachary Charles, Nicole Mitchell, Krishna Pillutla, Michael Reneer, and Zachary Garrett. Towards federated foundation models: Scalable dataset pipelines for group-structured learning. Thirty-seventh Conference on Neural Information Processing Systems Datasets and Benchmarks Track, 2023.

---

### Review · Reviewer_yAgt · 2023-12-13

**Summary Of Contributions:**

The paper introduces a novel algorithm FedFish for federated learning based on function space aggregation. In contrast to the standard approach of FedAvg that updates the global server model by minimizing the average distance between the parameters (which corresponds to taking an average of the client parameters), FedFish aims to minimize the average distance in function space betweeen the output functions. This is especially relevant for settings with high heterogeneity across clients since the FedAvg approach suffers from divergence of the model parametes as the number of local steps is increased.

FedFish considers the function space metric defined by the KL-divergence of the softmax outputs, averaged over training data. It then approximates this objective through a quadratic approximation defined by the Fisher Information matrix.

**Audience:**

Yes

**Broader Impact Concerns:**

The paper does not posit any broader impact concerns.

**Claims And Evidence:**

Yes

**Requested Changes:**

## Missing references that should be included for securing recommendation:

- A number of references related to the alleviation of ``client drift" are missing, such as:

     Dandi, Yatin, Luis Barba, and Martin Jaggi. "Implicit gradient alignment in distributed and federated learning." Proceedings of the AAAI Conference on Artificial Intelligence. Vol. 36. No. 6. 2022.

## Suggested changes that would strengthen the work:

- As an ablation study, it could be useful to include experiments utilizing the full Fisher-information matrix or other function space metrics on toy datasets.

**Strengths And Weaknesses:**

# Strengths:

- The paper is well written.
- The idea of aggregating models in the function space is well-motivated and novel.
- The paper contains well-designed experiments for both toy and real datasets to validate the claimed improvements, such as the change in the improvements for varying number of local steps.

# Weaknesses

- The paper lacks any theoretical analysis or guarantees.
- The ``function space" metric defined by the KL-divergence of the softmax outputs requires additional motivation. It should further be emphasized that the metric is limited to imposing similarity of the function outputs on the training data, not on the entire input space.
- The proposed approach isn't compared to algorithms such as SCAFFOLD (Karimireddy et al., 2020) and FedGA (Dandi et al., 2022) which also improve over FedAVG in the presence of client heterogeneity.

---

> ### Author Response · Authors · 2023-12-27
> **Response to Reviewer yAgt**
>
> We thank the reviewer for their summary of our work, thoughtful feedback and suggested changes, which are discussed below. We would like to reiterate that our paper makes primarily empirical contributions, motivated by the function space perspective and approximations to make it practical in realistic federated learning settings.
>
> > The ``function space" metric defined by the KL-divergence of the softmax outputs requires additional motivation. It should further be emphasized that the metric is limited to imposing similarity of the function outputs on the training data, not on the entire input space.
>
> The use of KL divergence in our derivations and experiments is motivated by the loss that client models are trained on, which is the cross-entropy loss for all our applications. Further, the function space metric is actually designed to impose similarity on the client training distributions since the function outputs of client models for inputs that are outside of their respective training distributions need not be meaningful. This is implemented in practice by optimizing our objective on the training data.
>
> > The proposed approach isn't compared to algorithms such as SCAFFOLD (Karimireddy et al., 2020) and FedGA (Dandi et al., 2022) which also improve over FedAVG in the presence of client heterogeneity.
>
> We agree with the reviewer that including comparisons to other methods aimed at addressing client data heterogeneity would strengthen the work. Given that the aim of our work is to strictly examine the aggregation strategy of federated algorithms, to this end, we chose FedAvg as a reasonable baseline and point of comparison.
>
> Other works, like FedGA (Dandi et al., 2022) and SCAFFOLD (Karimireddy et al., 2020), modify client optimization routines to deal with client heterogeneity. In contrast, FedFish modifies the aggregation of client models, which is complementary to and compatible with the above strategies. In fact, Fisher-weighting can be used as a replacement for simple parameter averaging in any of these settings.
>
> > Missing references that should be included for securing recommendation:
>
> Thank you for pointing out the reference [1]. We have included this citation, among the other relevant ones, in the Related Works section of the updated paper.
>
> > Suggested changes that would strengthen the work: As an ablation study, it could be useful to include experiments utilizing the full Fisher-information matrix or other function space metrics on toy datasets.
>
> While using the full Fisher Information matrix would be a useful ablation, it is challenging to implement even in toy settings without further approximations. With the full Fisher, the closed-form solution to our optimization problem would involve inverting a sum of Fisher Information matrices, which in practice need not be invertible. However, we refer the reviewer to our toy visualization in Figure 2 to compare FedAvg and FedFish with the true functions learned by client models (depicted as dotted curves).
>
> Thank you again for the suggestions to strengthen our work! We have updated our draft with the requested changes and welcome any further discussion.
>
> [1] Dandi, Yatin, Luis Barba, and Martin Jaggi. "Implicit gradient alignment in distributed and federated learning." Proceedings of the AAAI Conference on Artificial Intelligence. Vol. 36. No. 6. 2022.

---

### Decision · Action_Editor_EWkD · 2024-01-15

**Recommendation:** Accept with minor revision

**Comment:**

Strengths:
- Introduces a novel function-space aggregation method for federated learning (FedFish).
- Well-motivated and novel idea of aggregating models in function space.
- Extensive and comprehensive empirical studies demonstrating improvement over FedAvg.
- Clear and well-written paper with easy-to-follow presentation.
- FedFish is robust to longer local training and client data heterogeneity.
- Outperforms FedAvg in extreme cases like disjoint client data supports.
- Enables efficient personalization via local fine-tuning.

Weaknesses:
- Lack of theoretical analysis or guarantees for FedFish.
- The "Function space" metric and its limitations need further explanation.
- Potential lack of novelty due to similarities with existing work (e.g., [1]).
- Higher communication cost compared to FedAvg due to Fisher information calculation.
The possible relationship between FedFish's aggregation and loss surface sharpness needs exploration.
- Convergence rate comparison with FedAvg is missing.

Overall: The reviewers offer mixed opinions on the FedFish paper. While they acknowledge its strengths in novelty, empirical performance, and clear presentation, they also raise concerns about theoretical foundations, potential lack of novelty, and higher communication costs. Further work on theoretical analysis, addressing limitations of the metric, and comparisons with existing algorithms could strengthen the paper's contribution.

- Reviewers 1 and 3 suggest comparing FedFish to other relevant algorithms like SCAFFOLD.
- Reviewer 2 recommends discussing the reasons behind observed performance improvements.

The authors have already completed some of the requests of the reviewers; we are asking to satisfy as many as possible of these requests that do not violate the acceptance criteria of TMLR.

**Audience:**

The topic (FL, algorithms, alternative methods to existing ones for FL, etc) is definitely interesting to the ML audience.

**Claims And Evidence:**

Per TLMR guidance, there are 3 possible outcomes: Accept, Accept with minor revision, and Reject (see Editorial Policies).
As described in the Acceptance criteria page, acceptance decisions should be made based on two criteria:

- Are the claims made in the submission supported by accurate, convincing, and clear evidence?
- Would at least some individuals in TMLR's audience be interested in the findings of this paper?

Currently, it is clear from the reviews that both of these questions are answered affirmatively in favor of the paper. While some requests to add more material and results (e.g., theory) are definitely reasonable and desirable, it is against the rules of the TMLR to reject a paper for something that was not claimed by the authors.

That being said, and given the suggestions by the reviewers during the post-rebuttal period, we suggest Accept with minor changes.